

# Estimating the age of *Heliconius* butterflies from calibrated photographs

Denise Dalbosco Dell'Aglio[1,2], Derya Akkaynak[2], W. Owen McMillan[2] and Chris D. Jiggins[1,2]

[1] Department of Zoology, University of Cambridge, Cambridge, United Kingdom
[2] Smithsonian Tropical Research Institute, Panama City, Panama

## ABSTRACT

Mating behaviour and predation avoidance in *Heliconius* involve visual colour signals; however, there is considerable inter-individual phenotypic variation in the appearance of colours. In particular, the red pigment varies from bright crimson to faded red. It has been thought that this variation is primarily due to pigment fading with age, although this has not been explicitly tested. Previous studies have shown the importance of red patterns in mate choice and that birds and butterflies might perceive these small colour differences. Using digital photography and calibrated colour images, we investigated whether the hue variation in the forewing dorsal red band of *Heliconius melpomene rosina* corresponds with age. We found that the red hue and age were highly associated, suggesting that red colour can indeed be used as a proxy for age in the study of wild-caught butterflies.

## INTRODUCTION

Butterflies are some of the most colourful living animals and their bright wing colours have attracted the attention of scientists and artists alike. Multiple selective factors affect the evolution of butterfly wing colours, which might be tuned to the visual systems of potential mates and predators. This might be particularly true for the genus *Heliconius*, which exhibits conspicuous colours as a warning signal of toxicity (*Benson, 1972*; *Langham, 2004*), and find and choose mates based on these same colour signals that are involved in predator avoidance (*Jiggins et al., 2001*; *Estrada & Jiggins, 2008*).

Although mating behaviour and predation avoidance in *Heliconius* is highly linked to colour, previous research has shown that some *Heliconius* species exhibit considerable phenotypic variation in colour. Analysis of the colour patterns of two polymorphic mimic butterflies, *Heliconius numata* and the genus *Melinaea*, suggests that small differences in contrast can be perceived by butterflies and birds (*Llaurens, Joron & Théry, 2014*). More-over, variation of wing colour spectra between populations of *Heliconius timareta* indicates that their colours are locally adapted for mimicry in very precise ways (*Mérot et al., 2016*).

Most studies of colouration in *Heliconius* butterflies have focused on the genetic basis for colour variation (*Reed, Mcmillan & Nagy, 2008*; *Reed et al., 2011*; *Pardo-Diaz et al., 2012*). Convergent gene expression in *H. melpomene* and *H. erato* is associated with red

Corresponding author
Denise Dalbosco Dell'Aglio,
ddd23@cam.ac.uk

wing elements (*Pardo-Diaz & Jiggins, 2014*). Moreover, the *H. melpomene* gene responsible for red colour pattern is genetically linked to a preference for red (*Naisbit, Jiggins & Mallet, 2001*; *Merrill et al., 2011*). From a morphological point of view, wing scales have ultra-structural differences which are correlated with pigmentation (*Gilbert et al., 1988*; *Aymone, Valente & De Araújo, 2013*). Red/brown scales in *Heliconius* are pigmented with xanthommatin and dihydroxanthommatin and vary in colour from bright red to brown due to variations in the pigment oxidation state (*Gilbert et al., 1988*). However, there is also phenotypic variation in red among individuals that do not differ genetically, in wing regions with the same pigmentation and ultrastructure, with variation in colour from bright crimson to faded red.

It has been suggested that this variation in red is due to oxidation of the red dihydroxanthommatin pigment as individuals age (*Crane, 1954*; *Ehrlich & Gilbert, 1973*; *Gilbert et al., 1988*). Previous studies have taken advantage of this phenomenon to measure arbitrary age structure in *Heliconius* using wing condition such as wear, dull colours and scale loss (*Ehrlich & Gilbert, 1973*; *Walters et al., 2012*). We here investigated whether red colour can be used as a proxy for age in the study butterflies removing human vision bias. We used analysis based on digital photography to investigate the association between wing colouration and age, a methodology increasingly common in studies of animal coloration due to its high-end technology and affordability (*Stevens et al., 2007*; *Akkaynak et al., 2014*). Here, we created an unbiased methodology to quantify age based on the "redness" of the forewing red band in *Heliconius* butterflies.

## MATERIAL AND METHODS

To quantify changes in the dorsal forewing red band, we used a set of 55 *Heliconius melpomene rosina* Boisduval 1870 wings from Owen McMillan's collection in Gamboa, Panama, raised in insectaries and of known age (in days after emergence).

To objectively characterise colour, photographs of the wings were taken using an Olympus OM-D EM-1 digital camera with an Olympus Zuiko Digital ED 60 mm f/2.8 macro lens (Olympus, Center Valley, PA, USA). Forewing specimens were photographed against a Kodak R-27 Gray card, with Munsell 18% reflectance (Eastman Kodak, Rochester, NY, USA). Camera RAW images were converted to DNG format using the Adobe DNG Converter®, and white balanced and colour corrected according to equations (5)–(8) described in *Akkaynak et al. (2014)* using an Xrite Color Checker (Xrite, Grand Rapids, MI, USA). For illumination, a Bolt VM-110 LED macro ring light was used (Gradus Group LLC, New York, NY, USA). Image manipulations were done using custom scripts written in MATLAB® in wide gamut Kodak ProPhoto RGB colour space.

Following colour calibration, RGB images were projected to L\*a\*b\* colour space (*Wyszeck & Stiles, 1982*) and the a\* channel of each image was given a threshold at a\*>10 to segment the red patch automatically and obtain a binary mask. In the L\*a\*b\* colour space, the a\* channel takes on values between −128 and +128, and represents red–green opponency. The b\* channel also takes on values between −128 and +128, and represents yellow–green opponency. The L\* channel varies between 0 and 100, and is a measure of

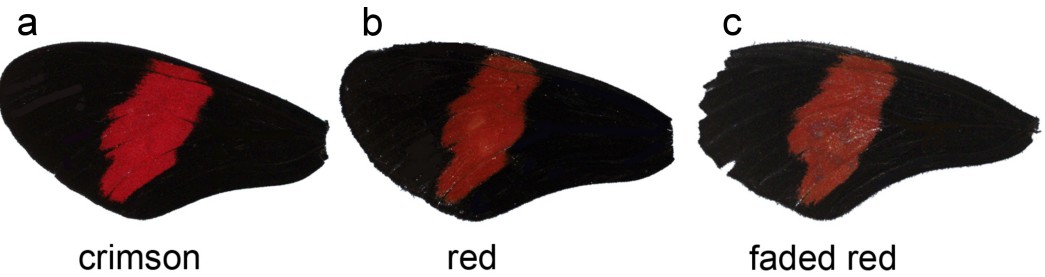

a        b        c

crimson      red      faded red

**Figure 1** **Forewings of *Heliconius melpomene rosina*.** Categories based on the appearance of colours to human vision: (A) crimson (fresh wings), (B) red (intermediate) and (C) faded red (worn).

lightness, with 100 being a bright white. Thus in this colour space, the more positive the a* value, the "redder" an image appears. The resulting binary mask was cleaned of isolated pixels using first morphological area opening, then closing. The automatically made masks were checked visually to ensure no pixels outside the red patch area or severe scale loss patches were included. The a* and b* values of the pixels inside the red patch area were averaged to obtain a representative colour for each specimen.

First, to test for differences between sexes, a $t$-test was performed. Next, to test for association between age and colour, a linear regression analysis was performed between age (days after emergence) and red measurement (*a). In addition, wings were sorted into groups using age/colour categories based on previous work in *H. ethilla* (*Ehrlich & Gilbert, 1973*). These subjective categories were named based on the appearance of the red band to human vision: crimson (fresh wings), red (intermediate) and faded red (worn) (Fig. 1). The categorical data was used to investigate the efficiency of human vision in categorizing subjective wing colouration.

## RESULTS AND DISCUSSION

Through calibrated digital images, we quantified the redness of *Heliconius* wings and found a strong association between age and fading of the colour (Fig. 2). We analysed 37 females and 18 males and found no significant difference in colour between the sexes ($t_{53,55} = 0.409, P = 0.684$), so both were combined in subsequent analyses. Our results showed that redness and age were highly associated ($t_{53,55} = -7.461, P < 0.001$), indicating that younger individuals have quantifiably "redder" patches. This suggests that the forewing dorsal red band changes colour with age as had been suggested previously (Fig. 2).

The samples included individuals across a wide range of age categories, reflecting the natural age structure of wild populations (*Ehrlich & Gilbert, 1973*). We have also shown that through calibrated photographs it is possible to distinguish more colour categories when compared to the limitations of the human vision. Although human visual categories do not fit especially well with the correlation (Fig. 2), the "crimson" category included all specimens that were less than 10 days old and "faded red" included specimens that were all older than 25 days (Fig. 2). Also, "faded red" might contain information about scale loss, in which was not explored in our methodology.
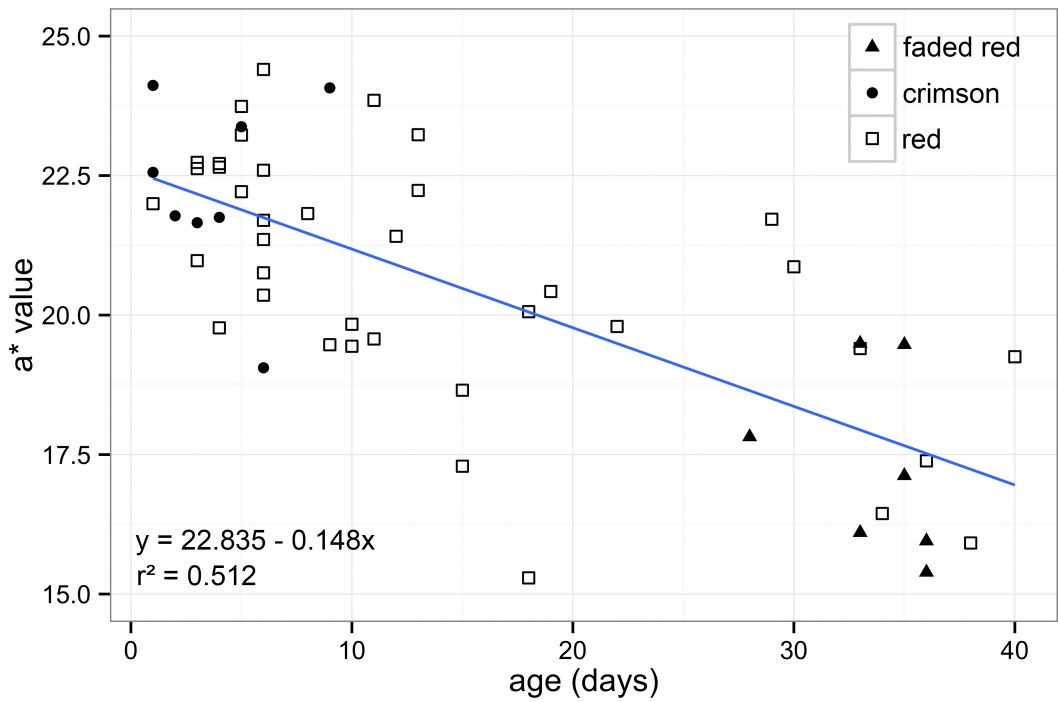

**Figure 2** **Forewing red band changes colour with age.** Association between redness (a* value) and age in days after emergence in *Heliconius melpomene rosina* forewing dorsal red band ($t_{53,55} = -7.461$, $P < 0.001$). Human visual categories: crimson (filled circles, $n = 9$), red (open squares, $n = 39$) and faded red (filled triangles, $n = 7$).

Age-dependent colour change has also been found in the tissues of a pentatomid bug, *Halyomorpha brevis*, with the change in red stronger in males (*Niva & Takeda, 2002*). Similar colour fading was found as a consequence of direct sun exposure in bees, *Bombus huntii*, which colour hue was correlated with wing wear and therefore could be a reliable measure of age (*Koch et al., 2014*). In butterfly wings, age-based colour fading has also been shown in *Colias eurytheme*, in which fading of structural colour was the most accurate predictor of male age. Females of this species choose their partners based on age, since new males produce more nutritious spermatophores such that colour might be a useful cue for mate choice (*Kemp, 2006*).

It is less clear whether there would be a similar benefit to such age discrimination in *Heliconius*. Females mate only once or a few times in their lifetime, depending on species, and the first mating occurs soon after eclosion (*Walters et al., 2012*). In contrast, males can mate throughout their life and there is no evidence that spermatophore quality is influenced by male age, although this has not been directly tested. Male choosiness in *Heliconius* is well documented (*Jiggins, Estrada & Rodrigues, 2004*; *Estrada & Jiggins, 2008*) as the spermatophore represents a considerable nutrient investment, providing the female with amino acids used in egg production (*Boggs, 1981*). Females are also likely to be choosy but this has been less well documented. Age, perhaps signalled by colour cues, might be a cue for mate choice in *Heliconius* and this would be interesting to test explicitly.

Furthermore, *H. erato* has red lateral filtering pigments which shift red receptor sensitivity, allowing butterflies to distinguish colours in the red–green spectrum with just a single LW-sensitive opsin (*Zaccardi et al., 2006*; *McCulloch, Osorio & Briscoe, 2016*). This means that *Heliconius* likely have better abilities to distinguish slight differences in the red colour range as compared to other nymphalids (*Zaccardi et al., 2006*). It would therefore be interesting to test whether adults can distinguish colours across the range demonstrated here among individuals of different ages. If age were an important trait in sexual selection, perhaps red filtering pigments are in part an adaptation for better mate discrimination in this range. It would be interesting to investigate how these colour differences would be seen through *Heliconius* vision.

It is also interesting to speculate about whether the fading of pigments might influence how predators perceive these butterflies. Although some degree of predator generalization is likely, it is also known that predators can distinguish fairly subtle differences in hue (*Langham, 2004*). Further experiments would be needed to determine whether colour fading might incur some cost in terms of increased predation. In conclusion, we have demonstrated how colour could be used to estimate age in population structure studies, and provided the groundwork for future studies of the fitness consequences of fading colours in *Heliconius* for mate choice and mimicry.

## ACKNOWLEDGEMENTS

We are very grateful to Adriana Tapia for her help with the butterfly collection, the Smithsonian Tropical Research Institute for logistical assistance, and three anonymous reviewers for valuable comments on the manuscript.

### Funding

This work was supported by the Cambridge Trust (UK) and CAPES (Brazil) to Denise Dalbosco Dell'Aglio, and the Smithsonian Tropical Research Institute (Panama) Short-term Fellowship to Derya Akkaynak. The funders had no role in study design, data collection and analysis, decision to publish, or preparation of the manuscript.

### Grant Disclosures

The following grant information was disclosed by the authors:
Cambridge Trust.
CAPES.
Smithsonian Tropical Research Institute.

### Competing Interests

The authors declare there are no competing interests.

## Author Contributions

- Denise Dalbosco Dell'Aglio conceived and designed the experiments, performed the experiments, analyzed the data, wrote the paper, prepared figures and/or tables, reviewed drafts of the paper.
- Derya Akkaynak conceived and designed the experiments, performed the experiments, analyzed the data, contributed reagents/materials/analysis tools, wrote the paper, reviewed drafts of the paper.
- W. Owen McMillan contributed reagents/materials/analysis tools, reviewed drafts of the paper.
- Chris D. Jiggins conceived and designed the experiments, wrote the paper, reviewed drafts of the paper.

## Data Availability

The code used is already published and its reference is included in the manuscript.

## Supplemental Information

Supplemental information for this article can be found online at http://dx.doi.org/10.7717/peerj.3821#supplemental-information.

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
