# Peer review of "Estimating the age of Heliconius butterflies from calibrated photographs"

_PeerJ, doi:10.7717/peerj.3821_

## Round 0.1 · original submission · Major Revisions

This is an interesting question and could result in a number of other studies. The reviewers raise a number of questions particularly regarding the methods and interpretation of the results that require comment before the manuscript can be accepted. However as I say I very much enjoyed the paper and what your are attempting to do.

·

Basic reporting

The article entitled “Estimating the age of Heliconius butterflies from calibrated photographs”, by Dell’Aglio and colleagues, investigates whether the hue variation in the forewing dorsal red band of a Heliconius butterfly species is associated with age. The authors shot digital photos from several wings dissected from vouchers with known biological ages, calibrated the images to avoid bias related to light conditions and scale loss, projected the images on color space, and extracted information of the red-green opponency to use as data in the analyses. The results indicate that redness of red band and butterfly age are negatively correlated, suggesting that a red fading in the wings of Heliconius melpome butterflies is expected when adults get older.

In general, the article is well written and have good quality data. The manuscript has a good discussion regarding the ecological and evolutionary implications of wing color changing in butterflies, and speculate about some studies that could be done in the future using the method presented in the paper, what I consider an inspiration for the scientific community. I really believe the method presented here represents an advance in the field.

I do not have many criticisms regarding the paper. The only two concerns I have are: (1) why have been stated that categorical data was used in the analysis if only qualitative data has been used, and (2) why authors make the statement that analysis was based in human visual perception if all data came from a machine assisted visual perception?

Experimental design

The experimental design is majorly ok, but the issues regarding (1) categorical vs. quantitative data analysis and (2) human vs. machine vision need further revision.

IIn page 4 (lines 83-84), authors made the following statement: “we used an analysis based on human visual perception of wing colour to investigate the correlation with the age”. When I read that, I thought there would be an analysis based on human subjective categorization of red fading (see in figure 1), but to my surprise all analyses were done using quantitative data obtained from computer processed images. In the Material and Methods section, authors have justified why they consider that human perception had been used, but the truth is that "naked human eye" has not been used to identify the association between "redness" and age. Instead, machine processed photos, shot under standard conditions and with no noise produced by scale loss, were examined. Although all imagens had been taken under standard white light and processed considering only visible light spectrum, using RGB channels, which we humans can see, no human eye is capable of precisely identify a hue value of redness and at the same time exclude information from scale loss. Thus, the way I see, at least we should consider that a “machine assisted vision” was used. In addition, I did not find any analysis concerning the three categories of red (crimson, red, fade red) as described in lines 90-97. The only reference to the subjective age/red categories is made on figure 2, where authors discriminate the red categories of all wings measured using a*value in the same plot. Even so, all data used in the statistics and discussed in the results section came only from quantitative data (a* values).

I think authors should consider that "real" human vision have not been used to identify the redness on the set of wings analyzed. Plus, I also recommend the authors to rethink about the necessity of include any reference to categorical data in the manuscript, since it not a major concern to answer the main question of the study (Is red fading of the wings associated with butterfly aging?). If authors choose to keep the references to the three classical age categories defined in the early 70's, I think it is needed to include another analysis comparing naked human eye efficiency in the identification of age classes using photos that have not been processed (scale loss edition).

Validity of the findings

The manuscript have good quality data, and unequivocally indicates that red fading in the wings of Heliconius melpomene is associated with aging of adults. The issues regarding human vs. machine vision, and categorical vs. quantitative data do not affect the validity of the findings. Although similar results are expected in other species of Heliconius, the results presented here refer only to one species of the genus. Perhaps should be important to make this clear in the title of the manuscript.

Additional comments

Although not discussed in the article, the results presented also raises important questions regarding the reliability of using only naked human eye and subjective categories based on color fading to identify the age of a butterfly. Figure 2 is very clear in showing that subjective red categories as a proxy for butterfly age are not good at all, since there is a massive overlap among categories. I think this issue could be included in the discussion (although not mandatory).

Reviewer 2 ·

Basic reporting

This study investigates whether the colour variation in the red ban exhibited by the butterfly Heliconius melpomene rosina is related to age. For this purpose the authors used digital photography and calibrated colour images of 55 individuals of this species whose age was known. The authors found a strong correlation between hue and age, according to human visual perception, so that younger individuals display a more intense red band (crimson) whereas older individuals present a faded red. The interpretation of these results is framed within literature on age-related colour fading, mate choice and possible relation with predator avoidance.


The figures are relevant for the paper, and the one related to the results is self-explanatory. The references cited are appropriate for the topic of the study. The paper is well written in general, although I have some suggestions for improvement.

Experimental design

I appreciate the information provided, considering that many of these assumptions (i.e. ‘redness’ is related to age) spread despite not having been tested. However, I think that the justification of your study needs to be reformulated. You say “Given the importance of red patterns in mate choice, we here investigated the influence of butterfly age on red colouration. We used an analysis based on human visual perception…”. If the rationale for your study is related to mate choice and, for example, whether or not age (through “redness”) could influence mate choice, using human visual perception is a strange choice. According to L164., vision has been studied in related species, so data should be available to give “butterfly values” to the differences in redness among individuals, according to your calibrated images. With the current knowledge on avian visual models, you could also check whether birds (potential predators) detect that variation in “redness”. However, with your choice of human perception these questions cannot be addressed. Thus, you need to re-formulate the justification of the study, placing more emphasis, for example, on why it would be useful for researchers to have a phenotypic indication of age. This would also make more ‘acceptable’ to include speculation in the discussion section, and would be a first step, as you imply, to address the relationship between colouration and age.

L114: The line about the statistical analysis is confusing. You state that you checked for a correlation between two variables using a linear regression, and then report a ‘c’ value in line 126… is this ‘r’ (from a correlation analysis)? A regression analysis is used when you want to predict a Y-value for a varying X-value, usually which is controlled by the researcher. I agree with your statement saying: “to test for a correlation….”, as this is what you can do to see to what extent, if at all, age and ‘redness’ are related. Also, you need to mention whether you did a parametric (Pearson) or a non-parametric (Spearman) correlation? Finally, you report no differences in colour between males and females with a t-test (L124). I would suggest you use a one-way ANOVA instead (I am assuming your colour data are normally distributed)?

L125-127: Probably the most important line in your paper, yet not clear in terms of the statistical approach. Is ‘c’ the correlation coefficient or an r2? If the former, that means you did a correlation analysis (which is, in my opinion, the right thing to do) rather than a linear regression. Much more accuracy about what you did is needed in the results section, even if I can see from the figure what the results are.

Validity of the findings

RESULTS ---> L125-127: Probably the most important line in your paper, yet not clear in terms of the statistical approach. Is ‘c’ the correlation coefficient or an r2? If the former, that means you did a correlation analysis (which is, in my opinion, the right thing to do) rather than a linear regression. Much more accuracy about what you did is needed in the results section, even if I can see from the figure what the results are.

Additional comments

See below a list of more specific comments.

L50: Please add references to support this statement!

L51: change ‘particular’ to ‘particularly’; add an ‘s’ at the end of ‘exhibit’, as here you are treating Heliconius as a genus (i.e., single)

L52: replace “that they are toxic” with “of toxicity”; also, insert a comma (,) after the brackets

L53: something is wrong here, I think? Do you mean “and find and choose mates based on the same colour signals that are involved in predator deterrence”? If I understand correctly, you want to say that the same colour signals are used to attract mates and repel predators… Right?

L58: ‘Melinaea’ should not start with capital letters

L62-63: change “for variation in colour” to “of colour variation”

L64: change ‘are’ to ‘is’, as you are referring to ‘gene expression’ (singular)

L73: insert a comma (,) after ‘genetically’

L78: … and predator avoidance(?)

L80: delete ‘the’ at the end of this line

L79-83. Consider splitting this line in two

L93: replace ‘captured’ with ‘taken’

L146: add ‘in’ before ‘which’

L162: remove the ‘s’ at the end of ‘shifts’

L163: add a comma (,) after sensitivity and replace ‘and allows’ with ‘allowing’

Reviewer 3 ·

Basic reporting

see the 'General comments for the author' below

Experimental design

see the 'General comments for the author' below

Validity of the findings

see the 'General comments for the author' below

Additional comments

The work reported here investigates whether the hue variation in the forewing dorsal red band corresponds to the age of the butterflies. Tested with 55 digital wing images, it is claimed that red colour can be used as indicator for age based on the strong (negative) correlation between age and a* values. As evident in Fig 2., all Crimson wings were <10 days and faded reds were about 30+ days old. So these two subjective colour labels could only put estimated age in these bands. The labelling of ‘red’ seems to cover a wide range (from 1-40 days) and hence not at all useful for age prediction.

On the other hand, a* values showed stronger correlation with age but this was not thoroughly investigated. Is this correlation value due to the distribution of the limited data in two fairly distinct clusters? There are very few samples in the age range 12<age<30. Hence the regression line is unlikely to be a good predictor of age. This could easily be tested by setting aside a subset of samples (as test set) from the regression analysis and the regression line obtained from the remaining data be then used to predict the age of the test set samples. Not being a bio-scientist/ecologist myself, I am not sure what sort of precision is generally expected or necessary in such age prediction exercise. But my personal feeling is that the proposed approach is not adequate in ‘estimating the age’ precisely enough. [I was curious to know if the two wings of the butterflies demonstrated similar colour values since they are the same age]. The authors may explore other colour representations as the discriminatory feature and preferably use a much larger data set to see if better age estimation is possible. Alternatively, if a rough estimation is good enough, this should be explicitly stated and then demonstrated how accurate are the age estimations under this definition.

The paper is generally well structured, language easy to read. Some (very few) typos still managed to get in which can easily be fixed.

---

## Round 0.2 · accepted · Accept

Thank you for considering the reviewers comments. I believe this has strengthened the paper and it is now acceptable for publication

Reviewer 2 ·

Basic reporting

I am happy with how the authors have addressed my comments from a previous version.

Experimental design

The authors have done the suggested changes to the manuscript in order to make their aims clear.

Validity of the findings

Ok

Additional comments

Thanks for having carefully addressed my previous comments. I have nothing left to add.